# Real-Time 3D Imaging and Inhibition Analysis of Various Amyloid Aggregations Using Quantum Dots

**DOI:** 10.3390/ijms21061978

**Published:** 2020-03-13

**Authors:** Xuguang Lin, Nuomin Galaqin, Reina Tainaka, Keiya Shimamori, Masahiro Kuragano, Taro Q. P. Noguchi, Kiyotaka Tokuraku

**Affiliations:** 1Department of Applied Science and Engineering, Muroran Institute of Technology, Muroran 050-8585, Japan; linxuguang4000@163.com (X.L.); 18041091@mmm.muroran-it.ac.jp (N.G.); teruteru_yrs718@yahoo.co.jp (R.T.); 19041040@mmm.muroran-it.ac.jp (K.S.); gano@mmm.muroran-it.ac.jp (M.K.); 2Department of Chemical Science and Engineering, National Institute of Technology, Miyakonojo College, Miyakonojo 885-8567, Japan; t-noguchi@cc.miyakonojo-nct.ac.jp

**Keywords:** amyloidosis, amyloid β, tau, α-synuclein, amyloid aggregation inhibitor, quantum dot, 4D-imaging

## Abstract

Amyloidosis refers to aggregates of protein that accumulate and are deposited as amyloid fibrils into plaques. When these are detected in organs, they are the main hallmark of Alzheimer’s disease, Parkinson’s disease, and other related diseases. Recent medical advances have shown that many precursors and proteins can induce amyloidosis even though the mechanism of amyloid aggregation and the relationship of these proteins to amyloidosis remains mostly unclear. In this study, we report the real-time 3D-imaging and inhibition analysis of amyloid β (Aβ), tau, and α-synuclein aggregation utilizing the affinity between quantum dots (QD) and amyloid aggregates. We successfully visualized these amyloid aggregations in real-time using fluorescence microscopy and confocal microscopy simply by adding commercially available QD. The observation by transmission electron microscopy (TEM) showed that QD particles bound to all amyloid fibrils. The 3D-imaging with QD revealed differences between amyloid aggregates composed of different amyloid peptides that could not be detected by TEM. We were also able to quantify the inhibition activities of these proteins by rosmarinic acid, which has high activity for Aβ aggregation, from fluorescence micrographs as half-maximal effective concentrations. These imaging techniques with QD serve as quick, easy, and powerful tools to understand amyloidosis and to discover drugs for therapies.

## 1. Introduction

Amyloidosis is a condition in which amyloid fibrils, as well as β-sheet structures, accumulate and are deposited extracellularly. Approximately 50 different peptides or proteins associated with amyloid diseases in humans or domestic animals have been reported worldwide [1,2,3,4]. Alzheimer’s disease (AD), Parkinson’s disease (PD), and other amyloidosis-related diseases are induced when misfolded Amyloid β (Aβ), tau, and α-synuclein proteins aggregate in the brain or in other organs and when hyperphosphorylated tau aggregates in neurofibrillary tangles [5,6,7,8,9]. The process of aggregation of these proteins is basically the same as that which forms amyloid fibrils rich in β-sheets. Furthermore, as these proteins aggregate, they can become generic toxins in higher organisms [10,11,12,13]. The aggregation of these proteins or peptides are a key step in the pathogenesis of amyloid diseases. Although cryo-electron microscopy has allowed the structure of these proteins to be elucidated and made available, the mechanism by which these proteins cause amyloid diseases is currently mostly unknown [14,15]. The diversity of amyloid formation may signal why amyloidosis is so difficult to treat with different clinical presentations, even when the same protein or peptide aggregates [16].

In amyloid disease, misfolded aggregation and accumulation of amyloid fibrils, when deposited in organs, constitutes a significant step towards the development of amyloidosis [2]. Various proteins or peptides serve as precursors for the formation of amyloid fibrils. Although effective therapy still currently remains out of reach [16], target therapy and plasma biomarkers are used to prevent and diagnose amyloidosis [17,18,19,20]. Therefore, it is important to find an effective compound or therapy to inhibit amyloid aggregation. Among recent research, none has been certified as medicine for the treatment of amyloidosis. Previously, we developed an Aβ-based real-time imaging method with quantum-dot (QD) nanoprobes, which have evolved as highly useful fluorescence probes in biological staining and diagnostics based on confocal and fluorescence microscopy over the past decade [21,22,23]. QDs are useful for long-term, single-molecule imaging in vitro. For these reasons, QDs could be an excellent tool for real-time monitoring of the aggregation of various amyloid proteins with nonspecific-binding-labeled fibrils. Utilizing this method, we developed a Microliter-Scale High-throughput Screening (MSHTS) system to screen Aβ aggregation inhibitors with QDs [21,24,25]. In this method (Appendix A), only 5-μL of a sample can be analyzed in a 1536-well plate, and the half-maximal effective concentration (EC_50_) is estimated as an inhibitory activity [25]. Furthermore, we successfully automated the MSHTS system using an auto-work station and demonstrated that this method could be applied to another protein, tau [21]. However, since the conventional MSHTS method requires labeling of the target protein by QDs, the issue of versatility remains unresolved. In the first report, we found that unlabeled QDs were able to bind to Aβ fibrils [25]. Therefore, in this study, to simplify the conventional MSHTS method, we attempted to visualize the 2D and 3D aggregation of various amyloid proteins using nonspecific binding. Consequently, we succeed in evaluating the aggregation inhibitory activity in each amyloid protein with the novel MSHTS method using nonspecific QD binding.

First, we compared the aggregation processes of Aβ, tau, and α-synuclein using unlabeled QDs and noted that the detailed aggregate structures of their proteins were not exactly the same. The results also showed that α-synuclein protein aggregation took a longer time to achieve than Aβ and tau. Using confocal microscopy and transmission electron microscopy (TEM), we detected details of their protein structure and analyzed amyloid fibrils after incubation with QD nanoprobes. As previously described [24], an inhibitory compound, rosmarinic acid (RA), was extracted from *Satureja hortensis* (summer savory), a spice belonging to the *Lamiaceae* family. RA has high Aβ aggregation inhibition activity, antioxidant properties, and can also inhibit xanthine oxidase [26]. An aggregation inhibitory test of these proteins using RA was performed, noting that RA inhibited the aggregation of Aβ but not tau. These results indicate that the modified MSHTS method using unlabeled QDs has the potential to be an easy and useful tool to search for aggregation inhibitors of various amyloids.

## 2. Results

### 2.1. Real-Time Imaging of Aggregation Processes of Various Amyloid Proteins

To observe the aggregation and fibrillization of Aβ, tau and α-synuclein proteins, we performed real-time imaging of each protein bound to the QDs via nonspecific binding using conventional fluorescence microscopy. Each protein sample was mixed with 30 nM Qdot^TM^ 605 ITK^TM^ amino (PEG) quantum dots (QD605) (Q21501MP; Thermo Fisher Scientific, Waltham, MA, USA) and incubated in a 1536-well plate at 37 °C. This imaging technique was able to directly monitor the physiological aggregation of each amyloid protein simply by adding commercially available QDs. We also confirmed that these amyloid aggregates can be visualized by QD655 (Q21521MP; Thermo Fisher Scientific) (data not shown). Aβ and tau aggregation can be visualized after incubation for 24 h, the ideal period reported in our previously study, which showed that tau protein aggregation was faster than Aβ aggregation [21] (Figure 1A, top and middle panels). Aggregation of α-synuclein was not observed by only incubating the monomer. Therefore, in this study, aggregation of α-synuclein was observed by adding α-synuclein aggregates that had been incubated at 37 °C for five days with stirring, as seeds which can promote α-synuclein protein aggregation (Appendix A). The α-synuclein aggregation process with 20% seed was observed for 168 h (Figure 1A, bottom panel). Aβ and tau completely aggregated within 24 h, whereas the α-synuclein protein needed more time to aggregate (168 h), even if the 20% seeds were added. These results indicate that the imaging technique using QDs can be applied to the aggregation of various proteins. In our previous study, we suggest that standard deviation (SD) values of fluorescence intensities of each pixel were correlated with the amount of amyloid protein aggregates [24]. As amyloid aggregations progressed, the variability of the SD values also increased. In this experiment, we compared the changes to SD values of Aβ, tau, and α-synuclein proteins during incubation (Figure 1B). The individual changes in the curves of the SD values of these three proteins were different. The SD values of Aβ and tau proteins increased more sharply than that of the α-synuclein protein. The SD value of the α-synuclein protein increased very slowly over time. The SD values of Aβ and tau peaked at around one day and that of α-synuclein at around one week. After these protein aggregations and their SD values peaked, the values no longer increased and plateaued.

### 2.2. 3D Observation of Aggregation of Various Amyloid Protein

Apart from observing 2D images, 3D aggregations of Aβ, tau, and α-synuclein proteins after incubation in the 1536-well plate with QD605 were directly observed in real time by confocal microscopy (Figure 2A). By using these QDs, we were able to distinguish the aggregate shapes of Aβ, tau, and α-synuclein, because the use of an imaging method using a QD without a drying step is suitable for detailed observations of aggregate shapes in solutions. The aggregation speed of each protein in the 3D-imaging was consistent with the 2D-imaging. As time goes on, the thickness of each protein increases due to aggregates. Tau showed the fastest aggregation that the aggregation of thickness can be observed in three h, while α-synuclein showed slower aggregation and showed the aggregation until 120 h. Furthermore, the XY view images showed that the size and density of each protein aggregation were different (Figure 2B). The increase in thickness of these protein aggregations after incubation can be observed in the XZ view (Figure 2A).

### 2.3. Transmission Electron Microscopy Observation of Various Amyloid Fibrils

TEM was used to confirm whether these amyloid proteins (Aβ, tau, and α-synuclein) formed fibrils and to assess whether QDs bound to amyloid fibrils (Figure 3). A low-magnitude image showed that these amyloid proteins had misfolded fibrils in vitro (Figure 3, top panel). A high magnitude image revealed that QDs had bonded evenly along the amyloid fiber (Figure 3, bottom panel). These results are consistent with our previous report [25]. However, these amyloid fibrils are difficult to distinguish, because almost all of them show a linear morphology with a cross structure and irregular length. These fibers that formed can cause amyloidosis and organ dysfunction when they are deposited and accumulate in organs or tissues. These results suggest that QD-based imaging can analyze three-dimensional amyloid aggregation and deposition in vivo in more detail than TEM imaging.

### 2.4. Effect of RA on the Aggregation of Three Amyloid Proteins

Using the MSHTS system, different concentrations of RA were used to evaluate the inhibitory activity of RA on the aggregation of three amyloid proteins (Aβ, tau, and α-synuclein). After incubation at 37 °C, images were captured by fluorescence microscopy (Figure 4A). Two-dimensional images were analyzed by ImageJ software to determine the SD value. The EC_50_ values of inhibitors were estimated from the inhibition curves (Figure 4B). The EC_50_ of Aβ, tau, and α-synuclein are 20.6 μM, ND, and 2.1 μM, respectively. RA inhibited the aggregation of Aβ and α-synuclein proteins in vitro in a dose-dependent manner, even though their EC_50_ values were different. However, tau aggregation was not completely inhibited, even when 1500-µM RA was used. Although tau aggregates were slightly reduced in the presence of 1500-µM RA, the EC_50_ could not be determined, because the SD value did not fall below 50%. These results indicate that the inhibitory activity of RA increased in the order of α-synuclein, Aβ, then tau.

## 3. Discussion

The global amyloidosis epidemic is a major complex combination of many diseases involving chronic inflammatory and misfolding of proteins that are characterized by the accumulation of amyloid-plaques and neurofibrillary tangles in tissues and organs [27,28,29]. Although there have been breakthroughs related to the structure and molecular mechanisms of amyloid proteins [30,31,32], the pathogenic mechanism of amyloidosis is still largely unknown. Over the years, therapy that directly targets amyloid aggregation and deposition in organs or tissues in order to clear it has so far been unsuccessful, and no approved treatment can revert or arrest the progression of this disease [7,33,34]. The procedure of amyloidosis in vivo is very sophisticated, and approximately 50 precursors (protein or peptides) have been reported, with some articles suggesting that pre-amyloid aggregates are the main cause of the induction of amyloidosis [35,36,37,38].

In this study, we attempted to elucidate the process of amyloid protein aggregation and assess the effects of RA, an aggregation inhibitor, by using a QD-based imaging and MSHTS system that employs fluorescence and confocal microscopy. In this procedure, we used commercially available QDs without any protein-labeling. The MSHTS system can be analyzed with a 5-µL sample volume when a 1536-well plate is used, and inhibitory activity can be estimated as EC_50_. QDs bound to amyloid fibrils, and the intervening space became dark, allowing images of the aggregates to be caught by fluorescence microscopy. The micrographs showed real-time aggregation, and the SD values also increased in a real-time manner. The real-time graphs (Figure 1B) showed a typical kinetic curve for amyloid aggregation that consisted of time lag, growth, and steady state phases, similar to recent 3D volume data that was obtained by confocal microscopy [21,25]. The lag time of Tau aggregation was probably too short to be detected under the MSHTS system. Since the time-dependent data revealed that the aggregation reached a plateau around a certain time, which was from 0 h until the amount of aggregation reaches saturation, the incubation period was fixed to that plateau time in the following screening steps.

Herein, we successful elucidated the images of real-time aggregation of three amyloid proteins (Aβ, tau, and α-synuclein) using the MSHTS system coupled to QD nanoprobes [21,24,25]. After these amyloid proteins were incubated in a 1536-well plate at 37 °C, their aggregation over time could be observed by confocal and fluorescence microscopy (Figure 1 and Figure 2). The accompanying SD values also increased concomitantly over the same period of time. The real-time 2D and 3D images showed that Aβ, tau, and α-synuclein aggregation occurred by 24 h, 24 h, and 168 h, respectively, while SD values reached saturation. In particular, the tau protein began to aggregate earlier than Aβ and α-synuclein. Moreover, 2D slice images from confocal microscopy (Figure 2) showed that the shapes of aggregates of these proteins were different, but the difference was not significant when observed by TEM (Figure 3).

We also analyzed the structural details of these amyloid protein aggregates by confocal microscopy. An important property of these proteins is that their aggregation structures are very similar, gathering in a spiral-like manner and exhibiting a mesh-like structure without the need for any incubation period. These amyloid fibril aggregations that form in vivo and that are deposited in organs or tissues can induce amyloidosis [39,40,41]. Based on previous data, we demonstrated that the aggregation and formation of amyloid proteins such as Aβ, tau, and α-synuclein are almost similar. Three-dimensional aspects related to the detailed structure of amyloid protein aggregation are highly nuanced, and TEM results reveal that amyloid fibrils had a linear morphology with cross structures and irregular length in vitro. The same fibrils can be extracted from diseased tissues or organs, although it cannot be proved whether these amyloid protein aggregation mechanisms are the same. Differences between each protein fiber are difficult to see with the naked eye [21]. Analysis of 3D images of aggregates using QDs may provide new information about the aggregation of different amyloid proteins.

In our previous researches [21,24], we found that the EC_50_ were different between the MSHTS system and thioflavin-T (ThT) assay in the Aβ protein, in which the EC_50_ of the MSHTS system was higher than ThT, because ThT has the potent of false positive effects in inner filter effects [42]. By contrast, the inner filter effects of the MSHTS system were smaller than ThT, which it adopts as a longer excitation spectrum and quantification from variability data of fluorescence intensity. In this study, we used RA to inhibit these amyloid protein aggregations based on the MSHTS system and calculated their EC_50_ values. Our data demonstrates that RA has high inhibitory activity of the aggregation in vitro of Aβ and α-synuclein. The aggregation inhibitory activity of RA on Aβ and tau was similar to the results using QD-labeled with these peptides [21,24]. This result demonstrated that the aggregation inhibitory activity could be evaluated using a commercially available QD that was not labeled with any peptides.

Although enormous efforts have been made by many researchers, currently there are no effective disease-modifying therapies available. Mutations of the amyloid precursor and some external factors, such as heating, create large challenges for the treatment of amyloidosis [43,44]. At present, some proposals and research involve treatments for amyloidosis based on antibodies, metal ions, and RNA interference, but none have been certified as medicine for amyloidosis therapy and applied clinically [17,45,46,47,48]. In this study, we elucidated 2D and 3D aggregation of three amyloid proteins (Aβ, tau, and α-synuclein) using QD, which has ability to directly observe the processes of aggregation and inhibition of these amyloid proteins in vitro. These aggregation processes, which involve proteins misfolding in human and animal diseases, are basically the same for these three amyloid proteins. Although these amyloid fibrils are polymorphic and consist of similarly structured proto-filaments in different species or in vivo and in vitro, amyloid proteins adopt highly similar β-arch conformations [15,49]. We used RA to inhibit amyloid protein (Aβ and α-synuclein) aggregation, showing a high inhibitory activity, but RA had no inhibitory effect on the tau protein. Our findings will be helpful to better understand the pathogenesis of amyloidosis, especially the progression of this disease. With more testing, over time, RA could be used for amyloidosis treatment as an effective medicine. It is our hope that real-time imaging using QDs may serve as an easy and useful tool in the future for the analysis of protein aggregation and to better understand amyloid proteins.

## 4. Materials and Methods

### 4.1. Materials

Human amyloid peptide of Aβ_42_ (4349-v) and QD605 (Q21501MP) were purchased from the Peptide Institute and Thermo Fisher Scientific, respectively. Mouse tau MBD fragments and α-synuclein were prepared according to the method described in 5.2. Prepared proteins were stored at −80 °C until use.

### 4.2. Preparation of tau and α-synuclein

The bacterial expression and purification of the tau MBD fragment was carried out as described previously [25,50]. Briefly, the expression plasmids were transformed into *Escherichia coli* (Rosetta™ (DE3) pLys, Burlington, MA, USA), and plasmid expression was induced by 1-mM isopropyl-1-thio-β-D-galactopyranoside. The heat-stable fraction of each extract was subjected to successive column chromatographies using a Bio-Scale^TM^ Mini UNOsphere^TM^ S (Bio-rad, Hercules, CA, USA) and a TOYOPEARL^®^ butyl column (Tosoh, Tokyo, Japan). Protein concentration was estimated using the method described by Lowry et al. (1951) [51], using bovine serum albumin (BSA) as the standard. Sodium dodecyl sulfate-polyacrylamide gel electrophoresis (SDS-PAGE) was carried out according to the method of Laemmli et al. (1970) [52].

The preparation of α-synuclein was performed as described previously with some modifications [50]. Briefly, Rosetta (DE3) cells (Novagen, Madison, MI, USA) harboring pT7-7 asyn WT plasmid (Addgene plasmid 36046; Watertown, MA, USA) were grown in a Luria-Bertani medium supplemented with 100-µg/mL ampicillin and 34-µg/mL chloramphenicol at 37 °C. When OD_600_ of the culture reached 0.6, the cells were induced by 0.5-mM isopropyl thio-β-D-galactoside for 3 h. Harvested cells were suspended in a buffer (10-mM Tris-HCl (pH 8.0), 1-mM EDTA, and 1-mM phenylmethylsulfonyl fluoride) and were then lysed by sonication for 10 s. After centrifugation at 10,000× *g*, the resultant supernatant was boiled for 20 min and recentrifuged at 16,000× *g*. Ammonium sulfate was gently added to the supernatant until it reached a saturation of 35%. After centrifugation at 13,500× *g*, additional ammonium sulfate was added until the supernatant reached a saturation of 50%. By further centrifuging at 13,500× *g*, the pellet was resuspended in 10-mM Tris-HCl (pH 7.4) and dialyzed overnight against 10-mM Tris-HCl (pH 7.4). The dialysate was applied to a HiTrap Q HP column (GE Healthcare Life Sciences, Buckinghamshire, UK), and the protein was eluted using a linear gradient of NaCl (0–500 mM) in 10-mM Tris-HCl (pH 7.4). The fraction containing α-synuclein was desalted using a PD Midi trap G-25 gel filtration column (GE Healthcare Life Sciences, Marlborough, MA, USA). The concentration of α-synuclein was determined using the extinction coefficient of 5600 M^−1^·cm^−2^. Purified α-synuclein was snap-frozen in liquid N_2_ and stored at −80 °C.

### 4.3. Microliter-Scale High-Throughput Screening System

The image was captured with an inverted fluorescence microscope (TE2000, Nikon, Tokyo, Japan) using a 4× objective equipped with a color CCD camera (DP72, Olympus, Tokyo, Japan) according to our previous study method of the MSHTS System [21]. Confocal laser microscopy aggregates in the 1536-well plate were observed by a confocal laser microscope (Nikon C2 Plus, Nikon, Tokyo, Japan) using a 20× objective. Amyloid protein aggregation can be visualized under a fluorescence microscope, and QD605, as well as proteins, were dispersed in samples, showing a red color. Furthermore, we analyzed details of the structure of amyloid aggregation based on confocal microscope imaging. The SD values were correlated with the amount of these amyloid protein aggregation when the thickness of the aggregate was within the range of the depth of focus. We selected 30-µM Aβ, 10-µM tau, and 6-µM α-synuclein as the concentrations, because a wide range of SD values is directly related to the sensitivity of this system.

### 4.4. Aggregation of Three Amyloid Proteins

In order to visualize α-synuclein aggregates with QD605, several solution conditions were reported in the article [53], but aggregations of the α-synuclein protein cannot observed by this imaging method using QDs. Thus, we tried to prepare seeds in which an α-synuclein monomer solution was stirred to make aggregates as seeds to promote the α-synuclein aggregation. A 20% seed concentration was the most efficient (Appendix A). The 400-μM α-synuclein solution was stirred at 37 °C for 5 days to prepare the seed for this experiment [54]. The α-synuclein sample (total of 20 μL) was prepared as follows: 10-μM α-synuclein with 20% seed and 30-nM QD605; Aβ and tau samples were prepared as described in our previous study [21]. Then, 5 μL of each sample was transferred into a 1536-well plate and centrifuged at 3700 rpm for 5 min at room temperature. After centrifugation, samples were incubated at 37 °C in an air incubator (SIB-35, Sansyo, Tokyo, Japan). Images of the aggregation were observed and captured with a color CCD camera (DP72, Olympus, Tokyo, Japan). Images of Aβ and tau samples were caught at 0 h, 3 h, 11 h, 15 h, and 24 h, while images of α-synuclein samples were caught every 24 h until 168 h. ImageJ software was used to analyze the SD value and draw a time-dependent curve of these amyloid proteins.

### 4.5. Transmission Electron Microscopy Observation

Aβ, tau, and α-synuclein samples were prepared in a total of 20 µL, the same volume as the MSHTS system, and incubated in PCR tubes at 37 °C. Samples were then deposited in 5-µL aliquots onto 200-mesh copper grids for 10 min and dried with filter paper, then washed with phosphate buffer saline (PBS). After washing, samples were negatively stained twice with 1% phosphotungstic acid for 10 min each time, then washed with PBS after each negative stain. Specimens were examined under an H-7600 transmission electron microscope (Hitachi) at 60 kV.

## 5. Conclusions

In summary, we successfully simplified a conventional MSHTS method using unlabeled QDs and were able to visualize the aggregation of three amyloid proteins: Aβ, tau, and α-synuclein in real time. Furthermore, we evaluated the aggregation inhibitory activity of RA in Aβ and α-synuclein amyloid proteins with the novel MSHTS method using nonspecific QD binding. We hope that this method of evaluation will contribute to the search and discovery of more aggregation inhibitors of various amyloid proteins.

## Figures and Tables

**Figure 1 ijms-21-01978-f001:**
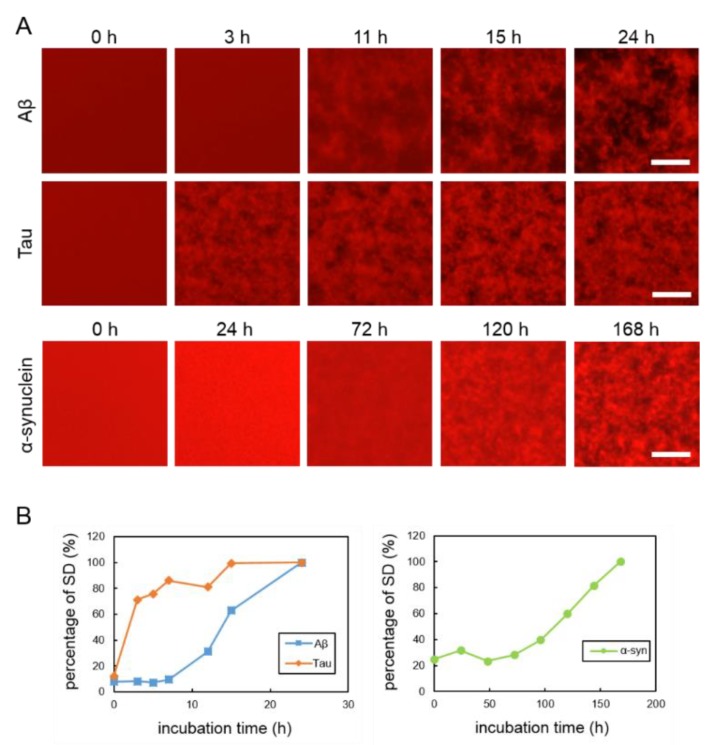
(**A**) Real-time imaging of aggregation processes of various amyloid proteins using QD605. Top: 30-µM Aβ, middle: 10-µM tau, and bottom: 10-µM α-synuclein with 20% seed. (**B**) Increase of SD values in each amyloid protein. SD values were determined by ImageJ software using the 2D images of Aβ, tau, and α-synuclein. Data represent the means from three independent samples. Scale bar = 100 µm.

**Figure 2 ijms-21-01978-f002:**
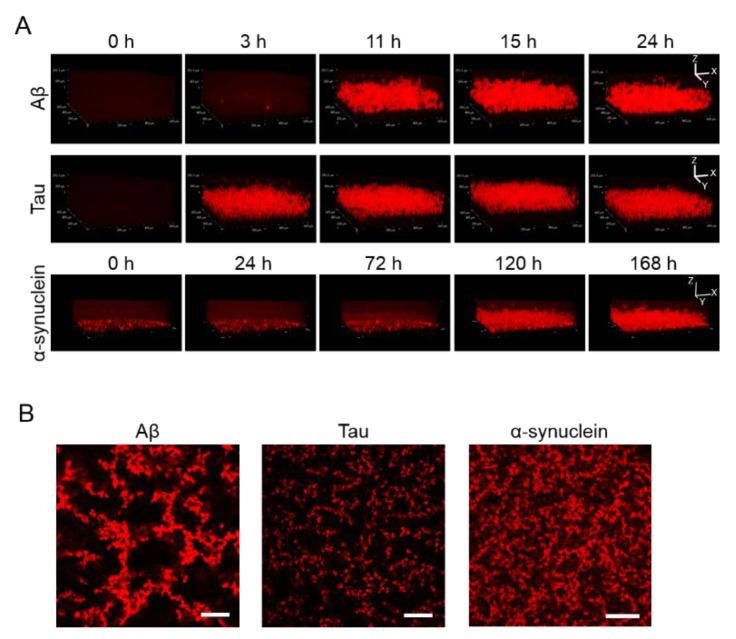
(**A**) 3D observation of the aggregation of Aβ, tau, and α-synuclein. Thirty-micrometer Aβ and 10-µΜ tau were incubated for 24 h. Ten-micrometer α-synuclein was incubated for 168 h. Sequential 3D reconstruction images of each time point are represented. (**B**) Image of the slices of aggregation of each protein in panel A. Note that aggregate sizes and densities are different among all three proteins. Scale bar = 100 µm.

**Figure 3 ijms-21-01978-f003:**
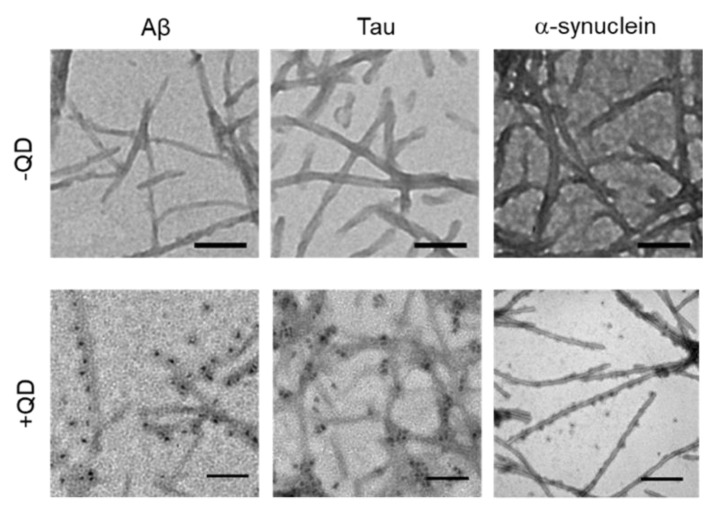
Transmission electron microscopy (TEM) observations of various amyloid fibrils of 30-µM Aβ, 10-µM tau, and 10-µM α-synuclein with quantum dot (QD) nanoprobes. Top panel, low-magnitude image of each protein without QDs. Bottom panel, high-magnitude images of each protein with 30-nM QDs. Note that QD nanoprobes, observed as black dots, bind to fibrils in each protein. Bars, 100 nm.

**Figure 4 ijms-21-01978-f004:**
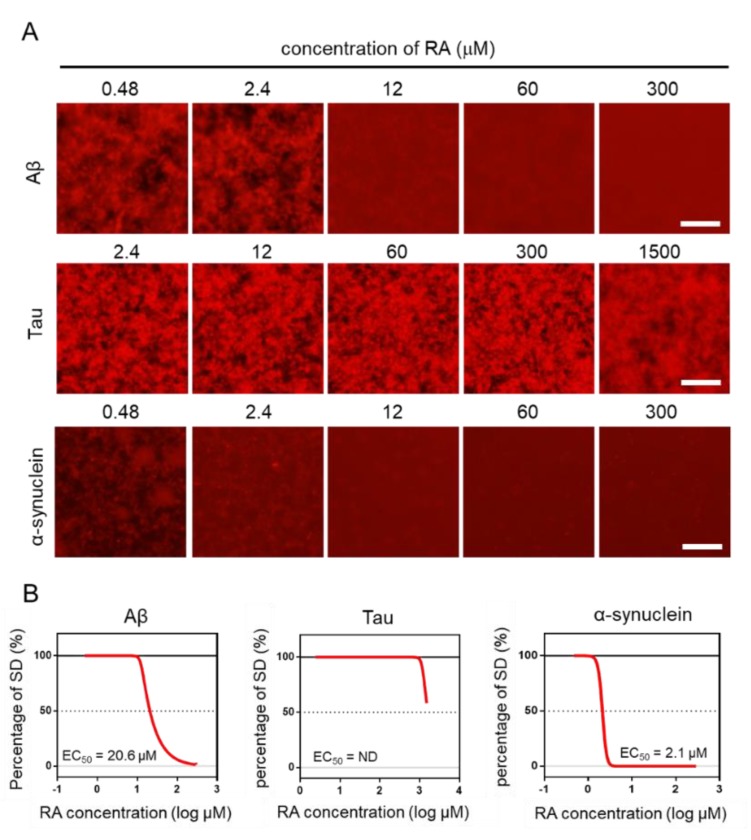
Estimation of EC_50_ by the Microliter-Scale High-throughput Screening (MSHTS) system. (**A**) Fluorescence image of each amyloid protein treated by rosmarinic acid (RA). (**B**) Aggregation inhibitory activity of RA for 30-µM Aβ, 10-µM tau, and 6-µM α-synuclein. The SD values from the fluorescence images were plotted against several concentrations of inhibitors. EC_50_ values were calculated from an inhibition curve (n = 3 independent experiments). Scale bar = 100 µm.

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
