# Peer review of "Real-Time 3D Imaging and Inhibition Analysis of Various Amyloid Aggregations Using Quantum Dots"

_ijms, 2020, doi:10.3390/ijms21061978_

Round 1

Reviewer 1 Report

In this manuscript, the authors describe a method and report the real-time 3D imaging and inhibition analysis of amyloid proteins utilising the affinity between quantum dots and amyloid aggregates. These authors have established a very promising method called Microliter-Scale High Throughput Screening (MSHTS). This technique has proved to be fast, efficient and relatively easy technique to monitor fibrillation and was also successfully used in the screening of inhibitory compounds. This present study is a follow up of papers published by the same group in particular a PloS one and more recently a Scientific Reports. Here, the authors improve their initial work by using commercial quantum dots and avoiding protein labelling increasing therefore the versatility and reducing the possible drawbacks coming from extra labels or Tags.
In my opinion, this manuscript deserves publication in IJMS because of this interesting and promising real-time imaging technique. However, I have few negative points that need to be answered.

Major Points:

- Figure 3 (upper panels):  Enlarge the fibrils obtained without QD in order to obtain comparable magnification.

- I miss some control experiments without QD using other techniques such as DLS or ThT fluorescence and a discussion part on how similar/different are the aggregation kinetics and EC50 with the ones obtained from SD analysis.

- Would it be possible to use this technique to visualize oligomers and to study the inhibiting effect on the most toxic species of amyloidogenesis. The authors do not need to write something about that in the MS but I would appreciate if they can answer my question. This is just pure curiosity.

Minor points:
- The authors need to edit slightly their final version in order to avoid some repeating parts.
- Line 149: "it IS difficult to distinguish the features…"

Author Response

Major Points:

- Figure 3 (upper panels):  Enlarge the fibrils obtained without QD in order to obtain comparable magnification.

Response 1: The fibril images without QD was enlarged to the same size as the fibril images with QD according to the comment of reviewer.

- I miss some control experiments without QD using other techniques such as DLS or ThT fluorescence and a discussion part on how similar/different are the aggregation kinetics and EC50 with the ones obtained from SD analysis.

Response 2: According to the reviewer's comment, we added text comparing the EC50 values obtained by this method and other methods to the discussion section (Line 226-230).

- Would it be possible to use this technique to visualize oligomers and to study the inhibiting effect on the most toxic species of amyloidogenesis. The authors do not need to write something about that in the MS but I would appreciate if they can answer my question. This is just pure curiosity.

Response 3: It is a most important trigger for amyloidosis that the toxicity of soluble oligomers aggregate. Therefore, inhibiting the aggregation of oligomers by certain compounds is also an effective treatment or prevention strategy for amyloidosis. And we have already proposed the quantification method for oligomerization using QDs [25]. In fact, we have confirmed that this technique can be used to evaluate oligomer formation inhibitory activity (manuscript in preparation).  

Minor points:

- The authors need to edit slightly their final version in order to avoid some repeating parts.

Response 4: We have simplified the redundant part according to the comment of other reviewer (Line 106-108).

- Line 149: "it IS difficult to distinguish the features…"

Response 5: We have changed the sentence a bit, keeping the assertion.

Reviewer 2 Report

The authors describe a method of imaging amyloid aggregations using commercially available quantum dots coupled to various microscopies to analyze 3D structures in real time. Applying this technology will provide a tool to search for aggregation inhibitors. This reviewer recommends publication as written. 

A couple of mistakes/typos were detected:

1.Line 57: Change "plasm" to "plasma"

2. Line 105-109: Two sentences are strung together that are redundant information.

3. Line 200: What "condition" are you referring to? This is not clear to the reader.

Author Response

1.Line 57: Change "plasm" to "plasma"

Response 1: "plasm" changed to "plasma".

2. Line 105-109: Two sentences are strung together that are redundant information.

Response 2: We have simplified the redundant information (Line 106-108).

3. Line 200: What "condition" are you referring to? This is not clear to the reader.

Response 3: We have changed the sentence more clearly (Line 198-199).

Round 2

Reviewer 1 Report

The manuscipt is now fine for publication in IJMS.